# Functional Blockage of S100A8/A9 Ameliorates Ischemia–Reperfusion Injury in the Lung

**DOI:** 10.3390/bioengineering9110673

**Published:** 2022-11-10

**Authors:** Kentaro Nakata, Mikio Okazaki, Tomohisa Sakaue, Rie Kinoshita, Yuhei Komoda, Dai Shimizu, Haruchika Yamamoto, Shin Tanaka, Ken Suzawa, Kazuhiko Shien, Kentaroh Miyoshi, Hiromasa Yamamoto, Toshiaki Ohara, Seiichiro Sugimoto, Masaomi Yamane, Akihiro Matsukawa, Masakiyo Sakaguchi, Shinichi Toyooka

**Affiliations:** 1Department of General Thoracic Surgery and Breast and Endocrinological Surgery, Okayama University Graduate School of Medicine, Dentistry and Pharmaceutical Sciences, Okayama 7008558, Japan; 2Department of Surgery, Division of Cardiovascular and Thoracic Surgery, Duke University School of Medicine, Durham, NC 27710, USA; 3Department of Cardiovascular and Thoracic Surgery, Ehime University Graduate School of Medicine, Shitsukawa, Toon 7910295, Ehime, Japan; 4Department of Cell Growth and Tumor Regulation, Proteo-Science Center (PROS), Ehime University, Shitsukawa, Toon 7910204, Ehime, Japan; 5Department of Cell Biology, Okayama University Graduate School of Medicine, Dentistry and Pharmaceutical Sciences, Okayama 7008558, Japan; 6Latner Thoracic Surgery Research Laboratories, Toronto General Hospital Research Institute, University Health Network, Toronto, ON M5G 2C4, Canada; 7Organ Transplant Center, Okayama University Hospital, Okayama 7008558, Japan; 8Department of Pathology and Experimental Medicine, Okayama University Graduate School of Medicine, Dentistry and Pharmaceutical Sciences, Okayama 7008558, Japan

**Keywords:** ischemia reperfusion injury, S100A8/A9, lung transplantation, damage-associated molecule patterns

## Abstract

(1) Background: Lung ischemia–reperfusion (IR) injury increases the mortality and morbidity of patients undergoing lung transplantation. The objective of this study was to identify the key initiator of lung IR injury and to evaluate pharmacological therapeutic approaches using a functional inhibitor against the identified molecule. (2) Methods: Using a mouse hilar clamp model, the combination of RNA sequencing and histological investigations revealed that neutrophil-derived S100A8/A9 plays a central role in inflammatory reactions during lung IR injury. Mice were assigned to sham and IR groups with or without the injection of anti-S100A8/A9 neutralizing monoclonal antibody (mAb). (3) Results: Anti-S100A8/A9 mAb treatment significantly attenuated plasma S100A8/A9 levels compared with control IgG. As evaluated by oxygenation capacity and neutrophil infiltration, the antibody treatment dramatically ameliorated the IR injury. The gene expression levels of cytokines and chemokines induced by IR injury were significantly reduced by the neutralizing antibody. Furthermore, the antibody treatment significantly reduced TUNEL-positive cells, indicating the presence of apoptotic cells. (4) Conclusions: We identified S100A8/A9 as a novel therapeutic target against lung IR injury.

## 1. Introduction

Lung transplantation shows the poorest outcomes among all solid organ transplantations, and most lung transplant recipients die or develop chronic lung allograft dysfunction (CLAD) within 4 years [1,2]. Lung ischemia–reperfusion (IR) injury induces primary graft dysfunction and increases the risk of CLAD development [3,4]. IR injury is caused by a sudden interruption of blood supply to an organ followed by resumption of blood flow. The initial tissue damage during hypoxia is enhanced by the subsequent resumption of blood supply and reoxygenation. IR injury results in acute cytotoxic stress, releasing acute stress molecules known as damage-associated molecular patterns (DAMPs), such as heat-shock proteins, high mobility group box 1, and S100A8/A9 [5,6]. DAMPs enhance the release of inflammatory cytokines that aggravate acute tissue damage. Furthermore, DAMPs are secreted from damaged cells [7].

The aim of this study was to identify the key initiator of lung IR injury, as the blockade of the early response after IR may be effective in ameliorating the inflammatory cycle mediated by damage-associated molecular patterns and inflammatory cytokines. RNA sequencing for lung tissue after IR revealed that S100A8/A9 mRNA levels showed the highest elevations with sufficient statistical significance at early stage after reperfusion. The combination of RNA sequencing and histological investigations revealed that neutrophil-derived S100A8/A9 is a key player in inflammatory reactions in lung IR injury. We hypothesized that S100A8/A9 can be a therapeutic target for lung IR injury. As S100A8/A9 is considered a key protein in lung cancer metastasis, previously, we developed an anti-S100A8/A9 neutralizing antibody that prevents lung cancer metastasis [8]. In addition, this antibody suppresses fibrosis by reducing inflammation in a mouse idiopathic pulmonary fibrosis model [9]. We used this antibody to demonstrate that the functional blockage of S100A8/A9 ameliorates lung IR injury.

## 2. Materials and Methods

### 2.1. Animals

Male 25–30 g C57BL/6J mice (CLEA Japan, Tokyo, Japan) were utilized for the experiments. All mice were handled according to the principles of laboratory animal care formulated by the National Society for Medical Research and the Guide for the Care and Use of Laboratory Animals. All experiments involving animals were performed in accordance with the ARRIVE guidelines [10].

### 2.2. Lung IR Model and Reagents

We prepared a left hilar clamp to generate an in vivo IR model as previously described [11]. The mice were fully anesthetized via the intraperitoneal administration of ketamine (0.1 mg/g) and xylazine (0.01 mg/g). The mice were connected to a VentElite small animal ventilator (Harvard Apparatus, Holliston, MA, USA) via orotracheal intubation with a 20G angiocatheter. Mechanical ventilation was kept at a tidal volume of 0.25 mL, a respiratory rate of 120/min, and general anesthesia with a combination of sevoflurane and 100% oxygen. The thoracic cavity was opened via left thoracotomy through the third intercostal space. For RNA sequencing and qRT-PCR of S100A8 and S100A9, mice with IR underwent left lung ischemia for 60 min, followed by 30, 60, and 120 min of reperfusion through the left hilar clamp using a microvascular clip (*n* = 3). For time-dependent analysis of the plasma and lung S100A8/A9 level, the mice were killed at 6 time points: before clamping, with 60 min ischemia (before reperfusion), followed by 10, 30, 60, and 120 min of reperfusion (*n* = 5). For the analysis of antibody treatment, 60 min of left hilar clamp was followed by 120 min of reperfusion on the animals receiving IR. Identical procedures were performed on the sham mice, but no hilar clamp was used. An anti-S100A8/A9 mAb was developed as reported before [8]. The mAbs against S100A8/A9 were purified from hybridoma and screened using cytokine and affinity assays. These assays were selected to suppress cell mobility in vitro and cancer metastasis in vivo. The mAb (clone 45) used in this experiment considerably suppresses lung inflammation by inhibiting the production of several proinflammatory cytokines in response to S100A8/A9. Mouse IgG (0107-01, Southern Biotech, Birmingham, AL, USA) was used as a control. Based on our previous protocol, anti-S100A8/A9 mAb (5 mg/kg) or control IgG (5 mg/kg) was administered intravenously via the tail vein 5 min before ischemia. An IgG-treated sham group (Sham-IgG), an anti-S100A8/A9 mAb-treated sham group (Sham-αS100), an IgG-treated IR group (IR-IgG), and an anti-S100A8/A9 mAb-treated IR group (IR-αS100) each included 5 mice. The body temperature of the mice was maintained at 37 ± 0.5 °C using a heating pad. After 120 min of reperfusion, to measure pulmonary dysfunction, left ventricular arterial blood was collected on FiO2 1.0 for quick blood gas analysis with the Rapid Lab 348 apparatus (Siemens Healthcare Diagnostics, Tokyo, Japan). The left lung was flash frozen with liquid nitrogen or preserved with 10% formaldehyde for biochemical and histological investigation, respectively.

### 2.3. Histological and Immunohistochemistry Staining

The 10% formalin-fixed lung tissue samples were embedded in paraffin. Tissue blocks were cut into 4 μm slices, put on glass slides, stained with hematoxylin and eosin, dehydrated, and mounted. Two pathologists analyzed the tissue samples using light microscopy in a blinded manner.

ImmPRESSTM (Vector Laboratories, Burlingame, CA, USA) was used for immunohistochemistry, and the procedure was carried out in accordance with the manufacturer’s instructions. Antigen retrieval was performed on the deparaffinized lung sections. The sections were blocked with 2.5% normal horse serum (Vector Laboratories) and treated at 37 °C for one hour with anti-S100A9 antibody (73425; Cell Signaling Technology, Danvers, MA, USA) or anti-Ly-6G antibody (87048; Cell Signaling Technology). The slides were mounted using the ImmPRESSTM Reagent Kit (Vector Laboratories). DAB detection and counterstaining were performed. The number of neutrophils in the lung was determined by counting Ly-6G-positive cells in three visual fields per segment at 200× magnification using ImageJ software version 1.53 (National Institutes of Health, Bethesda, MD, USA). The average number was then calculated. For immunofluorescence and costaining of S100A9 and MPO, the sections were probed with anti-S100A9 antibody (Cell Signaling Technology) and anti-MPO antibody (AF3667, R&D Systems, Minneapolis, MN, USA) for 12 h, followed by treatment with fluorescein isothiocyanate-AffiniPure donkey anti-goat IgG (H + L) antibody (Jackson ImmunoResearch Labs, West Grove, PA, USA) and Alexa Fluor 568 goat anti-rabbit IgG (H + L) (Invitrogen, Waltham, CA, USA). Fluorescence images were obtained using a Nikon A1 confocal laser-scanning microscope (Nikon Co., Ltd., Tokyo, Japan).

### 2.4. ELISA

The protein abundance of S100A8/A9, interleukin (IL)-1β, and IL-6 and tumor necrosis factor (TNF)-α in the plasma and lung tissue samples of mice were evaluated using Mouse S100A8/S100A9 Heterodimer, IL-1 beta/IL-1F2 and TNF-alpha DuoSet ELISA (DY8596-05, DY401-05, 406-05, 410-05, R&D Systems) according to the manufacturer’s instructions.

### 2.5. RNA Isolation and Sequencing

The total RNA was purified from the lung tissues using Isogen II (NipponGene, Tokyo, Japan) according to the manufacturer’s instructions. The RNA integrity number of the isolated RNAs was measured using a Bioanalyzer RNA6000 Nano kit. RNA (500 ng) was subjected to RNA sequencing using the NEBNext Ultra Directional RNA Library Prep Kit for Illumina and the NEBNext rRNA Depletion Kit (Human/Mouse/Rat) (New England Bio Labs, Cambridge, UK). Each library (19 pM) was sequenced using the Miseq Reagent V3 150 cycle kit on an MiSeq system (Illumina Inc., San Diego, CA, USA). Fragments per kilobase of exon per million fragments mapped reads data were used for relative expression analysis. Signal pathway analysis was performed using IPA software (Qiagen, Hilden, Germany).

### 2.6. Quantitative Real-Time Reverse Transcription Polymerase Chain Reaction (qRT-PCR)

The total RNA extracted from lung tissue was reverse-transcribed using a high-capacity cDNA reverse-transcription kit (Thermo Fisher Scientific, Waltham, MA, USA), according to the manufacturer’s instructions. qRT-PCR was performed using primers for genes encoding IL-1β, IL-6, IL-12, TNF-α, C-X-C motif chemokine ligand (CXCL)-1, CXCL-2, and GAPDH—Mm00434228_m1, Mm00446190_m1, Mm00434169_m1, Mm00443258_m1, Mm04207460_m1, Mm00436450_m1, and Mm99999915_g1, respectively. The cDNA and each primer and probe set (Thermo Fisher Scientific) were combined with TaqMan Fast Advanced Master Mix (Applied Biosystems, Waltham, MA, USA), and mRNA expression levels were measured using the ABI StepOnePlus Real-Time PCR Instrument (Thermo Fisher Scientific). To quantify the S100A8 and S100A9 mRNA level, a real-time PCR system (SYBR Green Master ROX; Roche Diagnostics, Basel, Switzerland) and primer pairs (5′-GGAAATCACCATGCCCTCTA [S100A8-f] and 5′-TGGCTGTCTTTGTGAGATGC [S100A8-r]) and (5′-GTCCAGGTCCTCCATGATGT [S100A9-f] and 5′-GAAGGAAGGACACCCTGACA [S100A9-r]) were used. GAPDH was adopted as a normalization control. Using the comparative delta-delta-CT method, the expression level relative to that of the samples of Sham-IgG was included as internal control in each real-time PCR experiment [12].

### 2.7. Western Blotting

Cryopreserved lung tissues were used for Western blotting. Western blot analysis was performed as we previously reported using following primary antibodies: p38, p-p38, JNK, pJNK, ERK, pERK (8690, 4511, 9252, 9251, 4695 and9101, respectively, Cell Signaling Technology), and beta-actin (MAB1501, Sigma-Aldrich, St. Louis, MO, USA) [11]. The secondary antibodies were anti-mouse or anti-rabbit immunoglobulin G (IgG) conjugated with horseradish peroxidase (Cell Signaling Technology). The membranes were examined using an enhanced chemiluminescence advanced Western blotting detection system (GE Healthcare, Piscataway, NJ, USA). Densitometry was assessed with ImageJ software (National Institutes of Health).

### 2.8. Apoptosis Analysis

We examined apoptosis in the lung tissues using a TUNEL assay, which was performed in accordance with the manufacturer’s guidelines (DeadEnd Fluorometric TUNEL System, Promega, Madison, WI, USA). Using ProLongTM Gold Antifade Mountant and DAPI (Thermo Fisher Scientific) stain, the slides were mounted. With a BZ-X810 fluorescent microscope (Keyence, Osaka, Japan), three random high-power fields at 20× magnification were analyzed for TUNEL-positive cells (green fluorescence) corresponding to nuclei (blue fluorescence). ImageJ (National Institutes of Health) was used to tally the number of apoptotic cells.

### 2.9. Statistical Analysis

Statistical analysis was performed using the Mann–Whitney U test for plasma and lung S100A8/A9 levels over time and one-way ANOVA for other parameters. All statistical analyses were performed using GraphPad Prism 8 software (GraphPad, Inc., San Diego, CA, USA) at a significance level of α = 0.05 (*p* < 0.05).

## 3. Results

### 3.1. S100A8/A9 as an Early-Response Gene of IR Injury in the Lung

The goal of our research was to identify genes that are initiators of IR injury and to develop functional inhibitors of the molecules. To identify the target genes for lung IR injury treatment, we investigated the mRNA expression profiles of lung tissues at 0, 30, 60, and 120 min after IR using RNA sequencing (Figure 1A). As shown in the volcano plot, the number of upregulated genes considerably increased over time (Figure 1B). Notably, S100A8 and S100A9 mRNA levels showed the highest elevations with sufficient statistical significance in the early stage after reperfusion. The S100A8 and S100A9 mRNA levels were significantly upregulated at 30 min and further increased within 120 min after IR compared with that at 0 min. In addition, among the S100 protein family members, S100A9 and S100A8 expression was particularly highly induced by IR injury in the lungs (Figure 1C). Quantitative real-time reverse-transcription polymerase chain reaction (qRT-PCR) of S100A8 and S100A9 showed that the mRNA levels of both genes after IR were elevated in a time-dependent manner (Figure 1D). These results indicate that S100A9 and S100A8, as early response genes, are therapeutic drug targets for acute pulmonary IR injury. The canonical pathway analysis using Ingenuity^®^ Pathway Analysis (IPA) software revealed that the enrichment of genes related to granulocyte adhesion and diapedesis significantly increased with the upregulations of S100A8 and S100A9 mRNA expression (Figure 2A). The signaling maps also clearly showed that genes related to tethering, rolling, and adhering granulocytes to the endothelial cells were strongly induced by IR injury in the lung (Appendix A). A signal network of gene expression profiles at 120 min after IR injury represented cytokine-induced acute inflammation responses, including granulocyte adhesion and diapedesis and S100A9 and S100A8 signaling (Figure 2B). These results suggest that granulocyte adhesion and induction of S100 genes might play critical roles as primers of IR injury (Figure 2C).

### 3.2. S100A8/A9 Production after IR Injury and Inhibition with Neutral Antibody

Enzyme-linked immunosorbent assay (ELISA) showed that the plasma S100A8/A9 level was significantly upregulated at 10 min after reperfusion compared to the level before clamp (*p* = 0.032). Although the plasma S100A8/A9 level was comparable at 10, 30, and 60 min after reperfusion, these levels were obviously upregulated at 120 min after reperfusion (*p* = 0.0079) (Figure 3A,B). The S100A8/A9 level in the lung tissue showed similar results: it was upregulated at 10 min after reperfusion compared to the level before clamp (*p* = 0.0079). It was also upregulated at 30, 60, and 120 min after reperfusion compared to the level before clamp (*p* = 0.0079, *p* = 0.016, and *p* = 0.0079, respectively) (Figure 3C).

The plasma S100A8/A9 level was significantly higher in the IR-IgG group than in the Sham-αS100A8/A9 group (*p* = 0.026). Prior administration of anti-S100A8/A9 monoclonal antibody (mAb) significantly decreased the plasma S100A8/A9 level after IR (*p* = 0.023) (Figure 3D,E) compared with control IgG. The expression of S100A8/A9 in the lung tissue samples did not significantly differ between the groups (Figure 3F).

### 3.3. S100A8/A9 Localization in IR Injury

We explored the localization of S100A8/A9 in the lung tissues after IR injury. To detect S100A8/A9, we used an S100A9 antibody, as almost all S100A8 and S100A9 molecules exist as S100A8/A9 heterodimers, which can be detected by the S100A9 antibody. Cells morphologically identified as neutrophils in hematoxylin and eosin (HE) staining and immunohistochemistry using S100A9 and myeloperoxidase (MPO) were distributed in approximately the same area (Figure 4A). Immunofluorescent histochemistry showed that S100A9 expression was limited to neutrophils detected by MPO (Figure 4B). These results of HE staining and immunofluorescent histochemistry indicate that most S100A8/A9 heterodimers localize in the neutrophils following lung IR injury.

### 3.4. Improvement in Lung Function Because of S100A8/A9 Inhibition

In order to determine the therapeutic effect of anti-S100A8/A9 mAb on pulmonary function, the results of arterial blood gas analysis of the left ventricle were assessed at 120 min after reperfusion (Figure 5A). IR-IgG group showed significant pulmonary dysfunction, as indicated by decreased partial pressure of O_2_ compared with that in the Sham-IgG and Sham-αS100A8/A9 groups (*p* = 0.022 and *p* = 0.022, respectively). Oxygenation was reduced in mice whose lungs were subjected to IR, and this reduced oxygenation was not observed in mice pretreated with S100A8/A9 (*p* = 0.026), indicating that anti-S100A8/A9 mAb improves lung function after IR.

### 3.5. Anti-S100A8/A9 mAb Reduces Lung Injury, S100A9-Positive Cells, and Neutrophil Infiltration

We performed HE staining of prepared specimens to determine whether anti-S100A8/A9 mAb could prevent lung injury (Figure 5B). The IR-IgG group was more severely impaired by IR injury than the Sham-IgG and Sham-αS100A8/A9 groups, as indicated by hemorrhagic congestion, interstitial oedema, neutrophilic infiltration, and septal thickening. These forms of damage were significantly improved by anti-S100A8/A9 mAb.

S100A9-positive cells were more abundant in the IR-IgG group than in the Sham-IgG (*p* = 0.0066) and Sham-αS100A8/A9 groups (*p* = 0.029). S100A9-positive cells were abased by anti-S100A8/A9 mAb following IR (*p* = 0.0029) (Figure 5C,D).

Neutrophils infiltrating lung tissue were also observed via immunohistochemistry with Ly-6G Ab (Figure 5E). Although the number of neutrophils in lung tissues was higher in the IR-IgG group compared with that in the sham groups (Sham-IgG, *p* = 0.00010; Sham-αS100A8/A9, *p* = 0.00020), there was a significantly smaller number of neutrophils that were seen in the lung tissues of the IR-αS100A8/A9 group than in thatof the IR-IgG group (*p* = 0.00010). Consistent with lung injury measurement, anti-S100A8/A9 mAb improved neutrophil infiltration after IR (Figure 5F).

### 3.6. Anti-S100A8/A9 mAb Suppresses Proinflammatory Cytokines and Chemokines

In order to analyze inflammation-related molecules secreted from damaged cells during IR, inflammatory cytokines and chemokines were investigated using qRT-PCR. The mRNA expression level of IL-1β, IL-6, TNF-α, CXCL-1 and CXCL-2 was significantly higher in the IR-IgG group than the sham group, whereas anti-S100A8/A9 mAb significantly reduced the expression of these proinflammatory mediators compared with the IR-IgG group (*p* = 0.00060, *p* = 0.00020, *p* = 0.0016, *p* = 0.046, and *p* = 0.0032, respectively) (Figure 6A). In addition, lung tissue protein levels of proinflammatory cytokines were evaluated using ELISA. Anti-S100A8/A9 mAb therapy reduced the elevated protein levels of IL-1β and TNF-α increased by IR injury (*p* < 0.0001, *p* = 0.00010) (Figure 6B).

### 3.7. Anti-S100A8/A9 mAb Ameliorated MAPK Signaling

We analyzed MAPKs, which are proinflammatory signals and downstream of multiligand receptors, including TLR4 and RAGE. IR injury significantly upregulated the phosphorylation of JNK, ERK and p38 compared with the sham group, and the antibody significantly reduced the phosphorylation of MAPKs. These data indicated that the antibody decreased activation of MAPK signaling (Figure 6C).

### 3.8. Anti-S100A8/A9 mAb Prevents Apoptosis in the Lung

The effect of anti-S100A8/A9 mAb on apoptosis was analyzed using the terminal deoxynucleotidyl transferase-mediated deoxyuridine triphosphate nick-end labeling (TUNEL) assay (Figure 7A). Although more apoptotic cells were shown in the IR-IgG group than the sham group, anti-S100A8/A9 mAb therapy significantly reduced apoptotic cells compared to the IR-IgG group (*p* = 0.0077) (Figure 7B).

## 4. Discussion

The objective of the present study was to identify genes that are initiators of lung IR injury and to develop functional inhibitors of the target molecules. We performed dynamic transcriptome analysis to identify the key initiator of lung IR injury. S100A8 and S100A9 mRNA levels showed the highest elevations with sufficient statistical significance at early stage after reperfusion, indicating that S100A9 and S100A8 might play a role as a master regulator for IR injury in lung. Furthermore, we determined whether S100A8/A9 is a central player in inflammatory reactions in lung IR injury and a potential therapeutic target using an anti-S100A8/A9 neutralizing antibody.

Early growth response 1 has been shown to be a trigger-switch transcription factor implicated in IR injury in the lung [13,14]. In the current study, basic transcriptomic analysis of the IR lung by RNA sequencing revealed that S100A8/A9 is an early regulator from the start of the reperfusion period. Consistent with our findings, in the myocardial IR injury model, S100A8/A9 was identified as a master regulator of cardiomyocyte death [15]. In lung transplantation, the expression of S100A8/A9 in bronchoalveolar lavage fluid was reportedly higher in patients with CLAD than in patients without CLAD [16,17]. However, the role of S100A8/A9 during lung IR injury remains unclear.

S100A8/A9 is a biological functional heterodimer consisting of two S100 family proteins—S100A8 and S100A9 [18]. Once the tissues are damaged by various causes, S100A8/A9 is secreted passively from neutrophils, macrophages, dendritic cells, and endothelial cells [19]. By binding to the receptor for advanced glycation end products (RAGE) and Toll-like receptor (TLR) 4, S100A8/A9 may function as an inflammatory mediator that promotes the recruitment of T cells, neutrophils, monocytes, and dendritic cells and the generation of inflammatory mediators. (Figure 2B) [7,20]. S100A8/A9 originates from neutrophils in IR injury and induces the secretion of neutrophils from the bone marrow to establish inflammatory responses. Although in the present study, the plasma S100A8/A9 protein levels were upregulated by IR, S100A8/A9 protein levels in the lung tissue were not upregulated by IR, indicating that therapeutic targets may be circulating S100A8/A9 rather than S100A8/A9 in the lung. However, mRNA levels of S100SA8/A9 in the lung were upregulated by IR. These findings suggest that S100A8/A9 were actively produced in the activated neutrophils in the lung, but S100A8/A9 were released from the lung and not remained in the lung. As expected, anti-S100A8/A9 antibody significantly reduced plasma S100A8/A9 levels, thereby ameliorating lung IR injury. Furthermore, anti-S100A8/A9 antibody decreased neutrophils within the IR-injured lung and S100A8/A9 was mainly upregulated within the neutrophils in the lung tissue. However, it may be because the reduction of circulating S100A8/A9 by antibody suppressed the inflammatory responses and consequently inhibited neutrophil infiltrations into the lung.

Although antibody treatment reduced protein level of S100A9 as shown by immunohistochemistry, the S100A8/A9 level in the lung tissue did not significantly differ between the groups, as determined by ELISA. We used the S100A9 antibody for immunohistochemistry and immunofluorescent histochemistry to detect S100A8/A9. This is because most S100A8 and S100A9 exist as S100A8/A9 heterodimer and S100A9 antibody has high detectivity [9]. In our previous study, we found that this mAb binds to S100A8/A9 to promote its degradation, but does not affect the results of ELISA [9]. This deviation in the results between immunohistochemistry and ELISA may be related to differences in the detection method.

As a part of the inflammatory reactions in IR injury, S100A8/A9 secreted from neutrophils promotes proinflammatory cytokines, including S100A8/A9 [21,22]. S100A8/A9 is a ligand for the TLRs and RAGE, causing upregulation of MAPK signaling, the NF-κB pathway, and inflammatory cytokines [21,23,24]. Anti-S100A8/A9 mAb therapy decreased the expression of MAPKs signaling and proinflammatory cytokines and chemokines, including IL-1, IL-6, TNF-, CXCL-1, and CXCL-2. Notably, CXCL-1 and CXCL-2 are important genes for neutrophil activation and migration in murine species [25]. Anti-S100A8/A9 mAb neutralizes S100A8/A9 secreted from neutrophils to regulate neutrophil activation and migration. Regulation of inflammatory cytokines and chemokines against lung IR injury has been reported to be an effective therapy [26,27]. Consistent with these previous reports, our results imply that treatment with anti-S100A8/A9 mAb alleviated lung IR injury by inhibiting the synthesis of inflammatory mediators. This marked improvement in lung IR injury may contribute to clinical lung transplantation [28,29].

A previous study using clinical samples in lung transplantation showed that apoptosis of cells induced by IR injury increases over time after reperfusion [30]. They showed that apoptosis was mainly observed in the alveolar epithelium, which is thought to be one of the causes of lung dysfunction. Although the role of apoptosis in IR injury is not fully understood, the suppression of apoptosis by a caspase inhibitor improved lung function after lung transplantation [31]. As with other apoptosis-inducing proteins, it has been shown that S100A8/A9 is essential in a vast array of mammalian cell types [32]. Furthermore, S100A8/A9 is known to work together with ROS to cause apoptosis [33]. We demonstrated that anti-S100A8/A9 mAb may inhibit IR-induced lung apoptosis. We have ensured the microscopic quantification of TUNEL positive cells by counting the costained areas with DAPI through automated counting from multiple locations on the specimen using ImageJ software. However, whether the inhibition of apoptosis occurred via direct inhibition by the antibody or through inhibition of IR injury remains unclear.

In the present study, we selected the hilar clamp model, which is a simpler and reliable model compared to the lung transplantation model, to minimize experimental errors. We plan to elucidate the possibility of using of this antibody in clinical settings by administering it to donors or recipients or by using the mouse lung transplant model containing perfusate that we previously developed [34]. We also need to plan to generate neutrophil-specific S100 knockout mice to further properly elucidate the mechanism by which the antibody suppressed IR injury. Furthermore, the anti-S100A8/A9 mAb used in this experiment can be mass produced. The next step toward clinical therapy requires testing of this drug in large animal lung transplantation models.

## 5. Conclusions

The combination of RNA sequencing and histological investigations revealed that neutrophil-derived S100A8/A9 plays a central role in inflammatory reactions in lung IR injury. Furthermore, the functional blockage of S100A8/A9 mAb by anti-S100A8/A9 mAb ameliorated lung IR injury by reducing neutrophil infiltration, inflammatory molecule expression, and apoptosis in the lung tissue. This antibody could be developed as a potential agent for the prevention and reduction of IR injury in clinical lung transplantation.

## Figures and Tables

**Figure 1 bioengineering-09-00673-f001:**
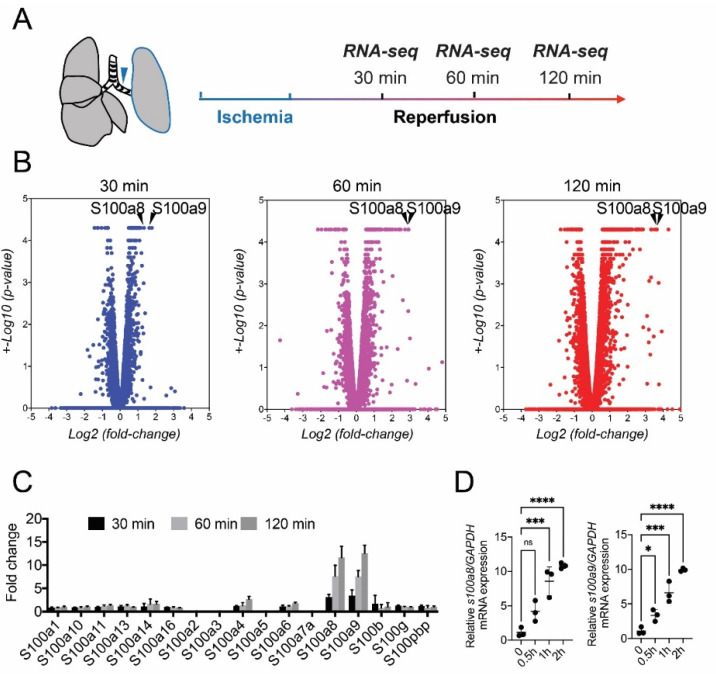
Experimental strategy for RNA sequencing. (**A**) After ischemia for 30 min, reperfusion was conducted for 30, 60, and 120 min, followed by RNA sequencing (*n* = 3). (**B**) Volcano plots of transcriptome data of whole lung tissue of ischemia-reperfusion (IR) mice (30 (blue), 60 (purple), and 120 min (red)). S100A8 and S100A9 are shown as a8 and a9, respectively. (**C**) Expression levels of S100 family genes in IR mice (IR-30, IR-60, and IR-120 min). All data were extracted from RNA sequencing data (fragments per kilobase of exon per million fragments mapped reads). (**D**) Black balls stand for relative expression levels of mRNAs encoding S100A8 and S100A9 by quantitative real-time reverse-transcription polymerase chain reaction (qRT-PCR) (*n* = 3). Data are expressed as means ± standard deviation (SD), * *p* < 0.05, *** *p* < 0.001, **** *p* < 0.0001.

**Figure 2 bioengineering-09-00673-f002:**
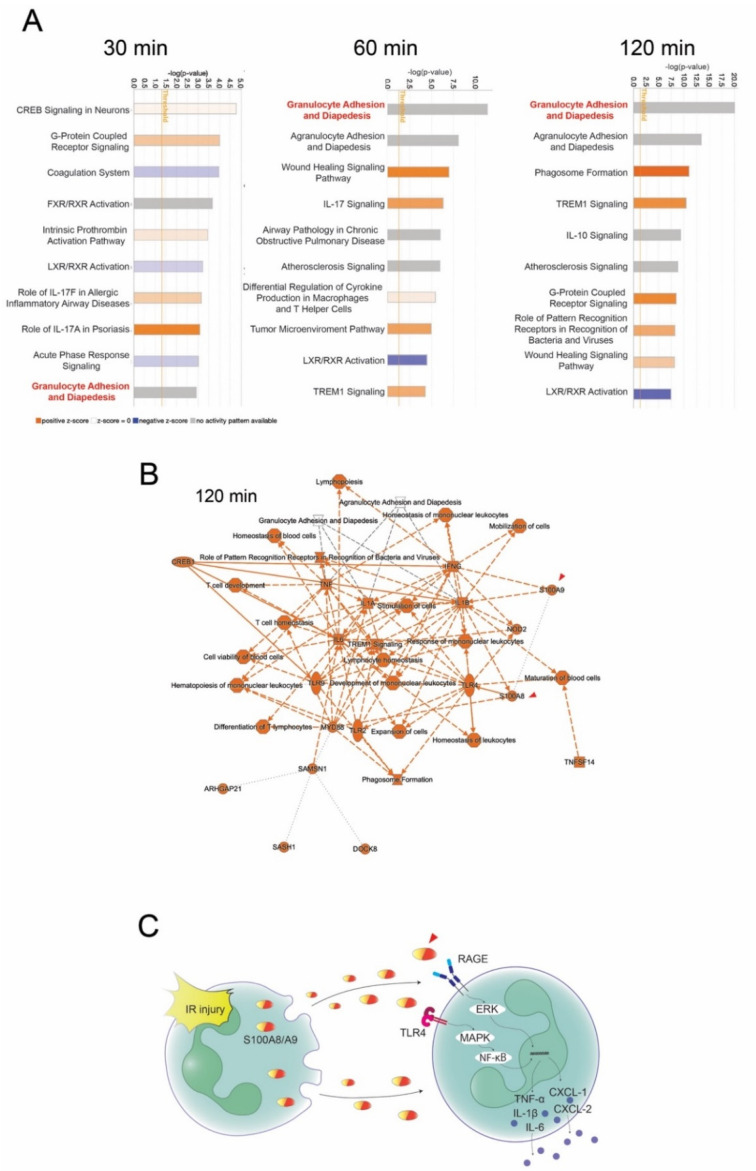
Gene enrichment analysis and signal maps of RNA sequencing data. (**A**) Ten enriched gene ontologies based on a canonical pathway database. Arrowheads indicate granulocyte adhesion and diapedesis signaling at 30, 60, and 120 min after IR injury. (**B**) A visual network of gene expression profiling at 120 min after IR. Arrowheads indicate the S100 genes. (**C**) A suggested schematic diagram of critical signal events of IR injury in the lung.

**Figure 3 bioengineering-09-00673-f003:**
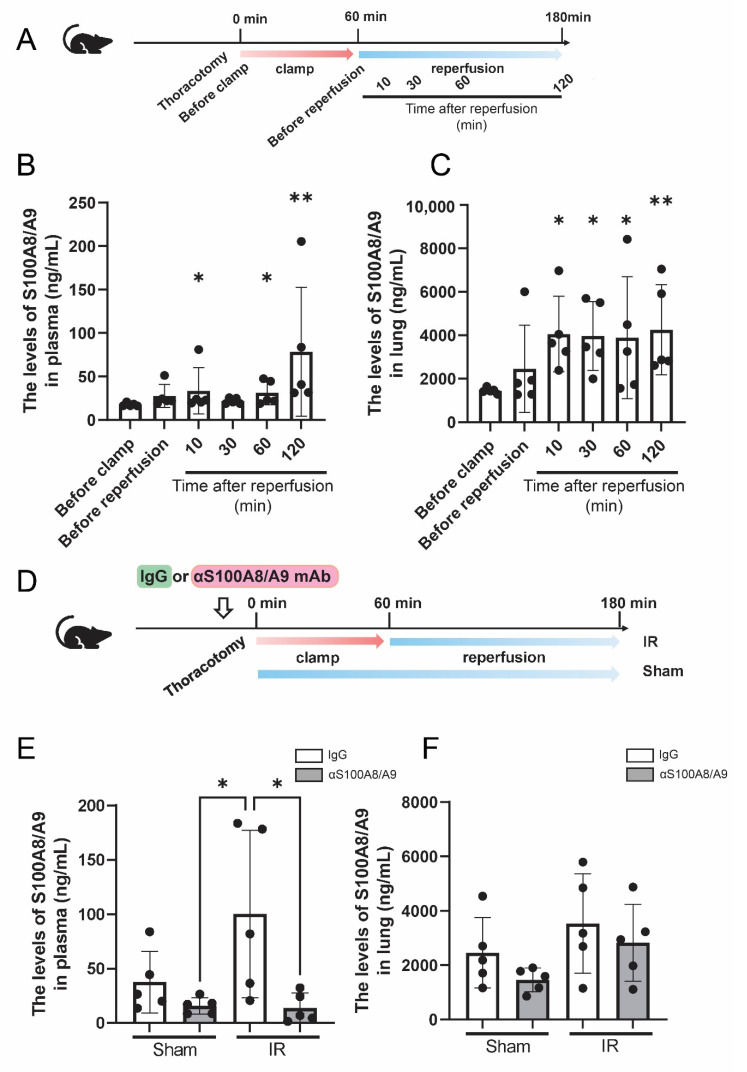
Dynamics of S100A8/A9 in the lung IR model. (**A**) Schematic diagram of time course experiment. (**B**) Changes in plasma concentration of S100A8/A9 before and after IR (*n* = 5, black ball). * *p* < 0.05, ** *p* < 0.01 compared with the before clamping group. (**C**) Changes in the S100A8/A9 level in the lung tissue before and after IR (*n* = 5, black ball). * *p* < 0.05, ** *p* < 0.01 compared with the before clamping group. (**D**) Schematic diagram of IR model. (**E**) Plasma level of S100A8/A9 at 120 min after reperfusion (*n* = 5, black ball). (**F**) S100A8/A9 levels in the lung tissue at 120 min after reperfusion (*n* = 5, black ball). Data are expressed as means ± SD.

**Figure 4 bioengineering-09-00673-f004:**
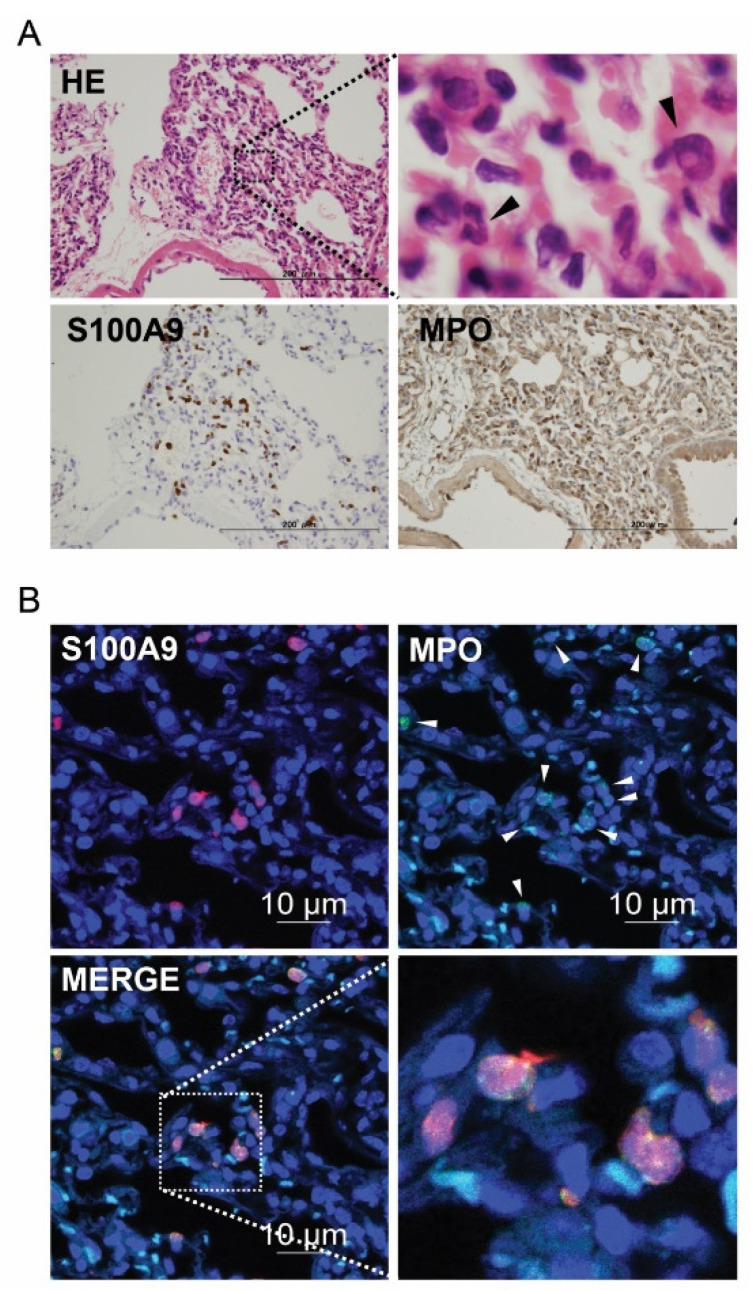
Localization of S100A8/A9 represented by S100A9-positive cells in the lung tissue under ischemia–reperfusion. (**A**) Hematoxylin and eosin staining and immunohistochemistry using S100A9 antibody and myeloperoxidase (MPO). Black arrowheads indicate neutrophils. (**B**) Images obtained using immunofluorescent chemistry; red indicates S100A9, green indicates MPO, and blue indicates DAPI. White arrowheads show neutrophils.

**Figure 5 bioengineering-09-00673-f005:**
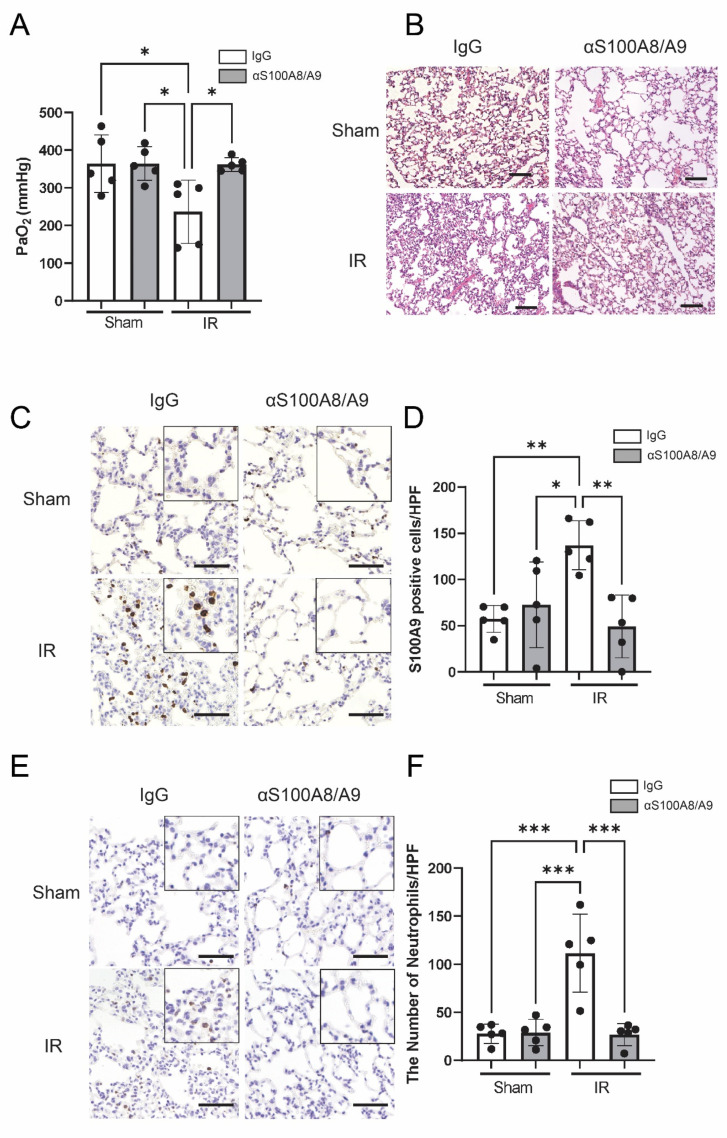
Oxygenation and pathogenesis show the therapeutic effect of anti-S100A8/A9 mAb on lung IR injury. (**A**) Lung oxygenation at 120 min following reperfusion (*n* = 5, black ball). (**B**) Hematoxylin and eosin staining of pulmonary parenchyma sections. Scale bars stands for 100 μm. (**C**) Immunohistochemistry of S100A9 to detect immune cells. (**D**) Number of S100A9-positive cells (*n* = 5, black ball). Scale bars stands for 50 μm. (**E**) Immunohistochemistry of neutrophils using the Ly-6G antibody. Scale bars stands for 50 μm. (**F**) Number of Ly-6G-positive neutrophils (*n* = 5, black ball). Data are expressed as means ± SD, * *p* < 0.05, ** *p* < 0.01, *** *p* < 0.001.

**Figure 6 bioengineering-09-00673-f006:**
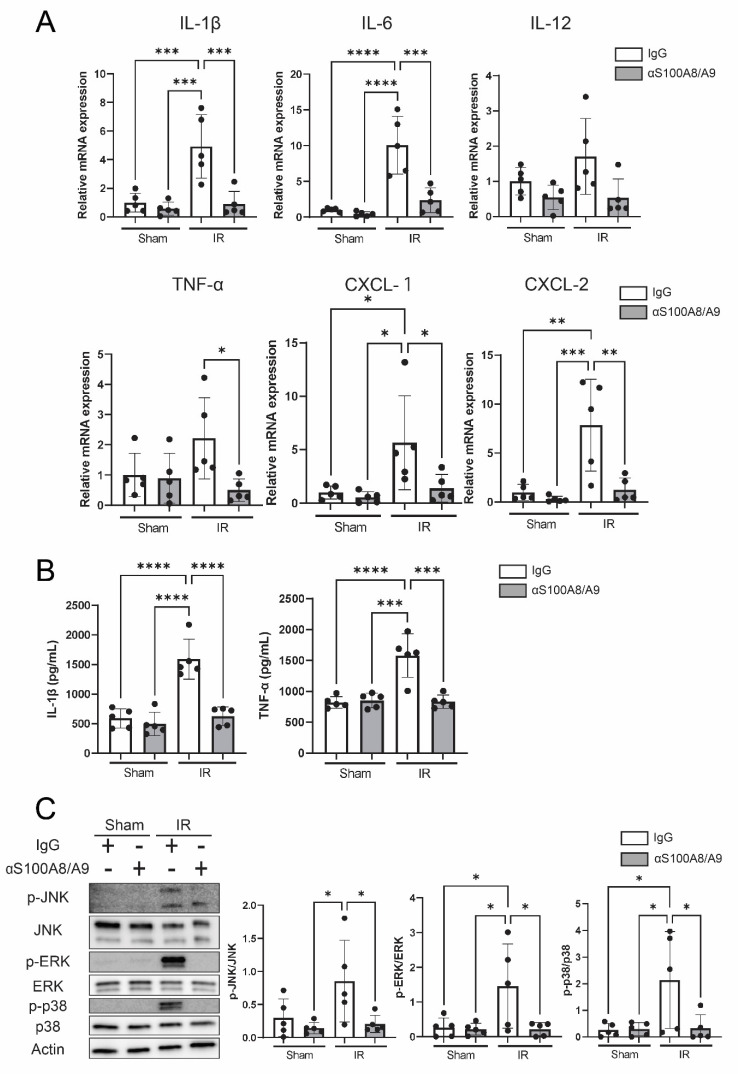
Anti-S100A8/A9 mAb inhibited inflammatory responses in IR injury. (**A**) Relative mRNA expression levels of inflammatory cytokines and chemokines were investigated by qRT-PCR (*n* = 5, black ball). (**B**) Protein levels of inflammatory cytokines in the lung were analyzed by ELISA (*n* = 5, black ball). (**C**) JNK, ERK and p38 and their phosphorylation were analyzed by Western blotting. Ratio of the phosphorylation was quantified by ImageJ (*n* = 5, black ball). Data are expressed as means ± SD, * *p* < 0.05, ** *p* < 0.01, *** *p* < 0.001, **** *p* < 0.0001.

**Figure 7 bioengineering-09-00673-f007:**
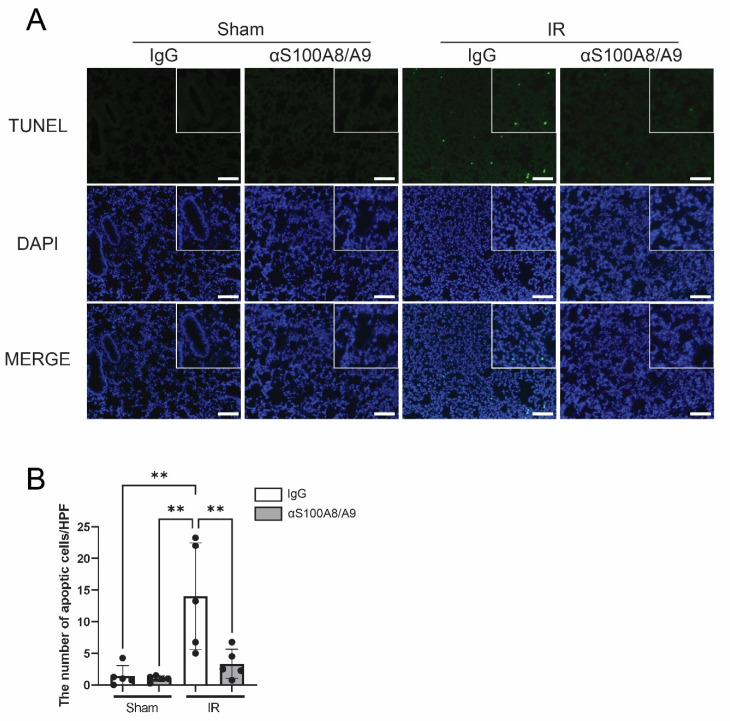
Anti-S100A8/A9 mAb inhibited apoptosis in the lung tissues during IR injury. (**A**) Terminal deoxynucleotidyl transferase-mediated deoxyuridine triphosphate nick-end labeling (TUNEL) immunofluorescence staining. Green staining indicates apoptotic cells. Scale bars stands for 50 μm. (**B**) Number of TUNEL-positive apoptotic cells (*n* = 5, black ball). Data are expressed as means ± SD, ** *p* < 0.01.

## Data Availability

The RNA-seq raw and processed data were deposited in the Gene Expression Omnibus (GEO) under accession GSE203238.

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
