# Peer review of "Functional Blockage of S100A8/A9 Ameliorates Ischemia–Reperfusion Injury in the Lung"

_bioengineering, 2022, doi:10.3390/bioengineering9110673_

Round 1

Reviewer 1 Report

General Summary

The authors investigated the key initiator of lung ischemia–reperfusion (IR) injury and evaluated pharmacological therapeutic approaches. They found that anti-S100A8/A9 mAb treatment significantly reduced plasma S100A8/A9 and cytokines and chemokines gene levels compared with control IgG and ameliorated the IR injury. The authors conclude that S100A8/A9 is a novel therapeutic target against lung IR injury.

General concerns:

1.     Please describe the rationale of an anti-S100A8/A9 mAb dose in this study.

2.     “Anti-S100A8/A9 mAb (5 mg/kg) or control IgG (5 mg/kg) was administered intravenously via the tail vein 5 min before ischemia”. “The plasma S100A8/A9 levels were obviously upregulated at 120 min after reperfusion (P = 0.0079) (Fig. 3A)”. Please explain the therapeutic effects of anti-S100A8/A9 mAb while anti-S100A8/A9 mAb was administered before the upregulation of S100A8/A9 plasma levels at 120 min after reperfusion.

Author Response

We wish to express our appreciation to the reviewers for their insightful comments on our paper. The comments have helped us significantly improve the paper.

Point 1: Please describe the rationale of an anti-S100A8/A9 mAb dose in this study.

Response 1: We appreciated the reviewer’s comment. In our previous study, we used this antibody and tried several doses using bleomycin induced pulmonary fibrosis mouse model. The dose used in the present study was the same as the optimal dose for inhibiting inflammation in the previous study. We added “Based on our previous protocol,” on line 96.

Araki, K.; Kinoshita, R.; Tomonobu, N.; Gohara, Y.; Tomida, S.; Takahashi, Y.; Senoo, S.; Taniguchi, A.; Itano, J.; Yamamoto, K. I.; Murata, H.; Suzawa, K.; Shien, K.; Yamamoto, H.; Okazaki, M.; Sugimoto, S.; Ichimura, K.; Nishibori, M.; Miyahara, N.; Toyooka, S.; Sakaguchi, M., The heterodimer S100A8/A9 is a potent therapeutic target for idiopathic pulmonary fibrosis. J Mol Med (Berl) 2021, 99 (1), 131-145.

Point 2.  “Anti-S100A8/A9 mAb (5 mg/kg) or control IgG (5 mg/kg) was administered intravenously via the tail vein 5 min before ischemia”. “The plasma S100A8/A9 levels were obviously upregulated at 120 min after reperfusion (P = 0.0079) (Fig. 3A)”. Please explain the therapeutic effects of anti-S100A8/A9 mAb while anti-S100A8/A9 mAb was administered before the upregulation of S100A8/A9 plasma levels at 120 min after reperfusion.

Response 2: Since plasma S100A8/A9 secreted from the lungs induces inflammation by cytotoxic stimuli, we administer anti-S100A8/A9 antibody before injury as the most reliable way to prevent plasma S100A8/A9 from increasing. Indeed, our most recent experiments have shown that these antibodies significantly inhibit neutrophil invasion in the lung tissue (figure 5EF), suggesting that preadministration of neutralizing antibody against S100A9 may effective for inflammation reactions after lung transplantation.

Reviewer 2 Report

I think it is a very interesting study with important implications in clinical practice if the results will be confirmed by other in vivo studies, as stated by the authors. Ischemia-reperfusion (IR) injury is a relevant clinical problem in lung transplantation, and the individuation of molecular paths leading to lung injury in this condition and consequent individuation of targeted therapies to prevent one of the causes of organ failure may have a great impact in these patients (doi: 10.3390/cells10040780).

Lack of regular perfusion has been reported as a cause of major complication also in other fields (doi: 10.1515/med-2019-0114)

The work is quite good but it needs several revisions:

- Line 84: animals were sacrificed before or after clamping?

- Line 312: instead of “ameliorated”, I would maybe better use “decreased”

- Do you think that lung ventilation with FiO2 100% during the ischemic phase could have increased lung damage in your animal model, since ROS eventually produced in lung tissue could not be washed out?

- Do you suggest that using anti-S100A8/A9 antibodies in lung perfusion solutions used during lung procurement could represent a valid therapeutic option? Or would it be more useful to use anti-S100A8/A9 antibodies before lung reperfusion in the transplant recipient?

- Do you think the use of these antibodies could also ameliorate lung function in lungs from DCD donors (in which classically the time of ischemia before reperfusion is longer)?

It would be kind to improve your manuscript as suggested.

Author Response

We wish to express our appreciation to the reviewers for their insightful comments on our paper. The comments have helped us significantly improve the paper.

Point 1: - Line 84: animals were sacrificed before or after clamping?

Response 1: We appreciated the reviewer’s comment.  For the time-dependent analysis of the plasma and lung S100A8/A9 level, the mice were sacrificed in 6 time points. It was difficult to understand as the reviewer pointed out, we added “in 6 time points” on line 86.

Point 2:  - Line 312: instead of “ameliorated”, I would maybe better use “decreased” 

Response 2: We use “decreased” instead of “ameliorated” on line 312.

Point 3.  - Do you think that lung ventilation with FiO2 100% during the ischemic phase could have increased lung damage in your animal model, since ROS eventually produced in lung tissue could not be washed out?

Response 3: As the reviewer pointed out, 100% oxygen would increase lung damage. The degree of IR injury can be adjusted by setting of oxygen concentration and ischemia time. However, we don't think there is a problem in determining the treatment effect if the control and treatment groups are in the same setting. Furthermore, the usage of 100% oxygen is the most easily reproducible setting and reduces experimental variability.

Point 4:  - Do you suggest that using anti-S100A8/A9 antibodies in lung perfusion solutions used during lung procurement could represent a valid therapeutic option? Or would it be more useful to use anti-S100A8/A9 antibodies before lung reperfusion in the transplant recipient?

Response 4: As the reviewer mentioned, the use of this antibody in the perfusate is a possible treatment and we would like to experiment with it. However, due to its inhibitory effect on neutrophil infiltration, we believe it is more likely to be more effective when used in the recipients than in the perfusate.

Point 5:  - Do you think the use of these antibodies could also ameliorate lung function in lungs from DCD donors (in which classically the time of ischemia before reperfusion is longer)?

Point 5: In lung transplantation from DCD, we believe that this antibody can ameliorate the damage of DCD lungs that are more severely damaged due to long warm ischemic time.

Reviewer 3 Report

Dear Authors,

It's a great focus on the subject. The research project was well designed.

The only suggestion is that flow cytometry is used to quantify apoptosis because microscopy is not quantifiable.

Clear and well-detailed discussions.

Congratulations, great work.

My Best Regards

Author Response

We wish to express our appreciation to the reviewers for their insightful comments on our paper. The comments have helped us significantly improve the paper.

Reviewer 4 Report

Reviewing the manuscript entitled, “Functional blockage of S100A8/A9 ameliorates ischemia–reper-2 fusion injury in the lung” by Nakata K er al., this is an article focusing on therapeutic effect of S100A8/A9 inhibition on pulmonary ischemia-reperfusion injury using a neutralizing antibody of S100A8/A9. Although this is an interesting manuscript for future transplant medicine, I think you need to fix some details. Therefore, the authors need to respond to the following concerns for acceptable quality.

The authors should provide a detailed explanation of how S100A8/A9 was targeted in this manuscript and the background of S100A8/A9 in the introduction section.

In 2.2. Lung IR model and reagents, the authors need to add a figure for your animal model for easy understanding the experiments or move the Figure 1A from the results section to 2.2.

In 2.6 Quantitative real-time reverse transcription polymerase chain reaction (qRT-PCR), I can't find any description about primers and measurement methods for S100A9 mRNA. The authors need to describe it.

Although line 195 to 196, you mentioned “In addition, among the S100 protein family members, S100A8 and S100A9 expression was specifically induced by IR injury in the lungs (Fig. 1C)”, what does “specifically” mean? If this indicate “in an idiosyncratic way”, it is out of line. The authors need to modify it.

In Figure 1B, What does a8 or a9 mean? The authors need to modify Figure 1 and its legend.

The authors need to change the graph of S100a9 and S100a8 in Figure 1D. I think the numerical order is easier to read.

At line 269 and Figure 5A, you mentioned “indicating that anti-S100A8/A9 269 mAb improves lung function after IR”. If you use the word “improvement”, you should show the data when it got worse.

The immunohistograms in Figures 5 and 7 are difficult to interpret. The authors need to modify them including their legends.

Although this manuscript abruptly showed the results of an apoptotic experiments, but I could not understand the purposefulness of carrying out these experiments. In this regard, the authors should discuss the relationship between S100A8/S100A9 functions and necrosis, apoptosis including necroptosis and pyroptosis as mechanisms of cell death in acute lung reperfusion in the discussion section.

Author Response

We wish to express our appreciation to the reviewers for their insightful comments on our paper. The comments have helped us significantly improve the paper.

Point 1: The authors should provide a detailed explanation of how S100A8/A9 was targeted in this manuscript and the background of S100A8/A9 in the introduction section.

Response 1: We appreciated the reviewer’s comment. S100A8/A9 was targeted because RNA sequencing revealed that S100A8/A9 mRNA levels showed the highest elevations with sufficient statistical significance at early stage after reperfusion. We added this sentence in the introduction section (line 56-58).

Point 2: In 2.2. Lung IR model and reagents, the authors need to add a figure for your animal model for easy understanding the experiments or move the Figure 1A from the results section to 2.2.

Response 2: We agree with the reviewer’s comment. We moved the Figure 1 to 2.2.

Point 3: In 2.6 Quantitative real-time reverse transcription polymerase chain reaction (qRT-PCR), I can't find any description about primers and measurement methods for S100A9 mRNA. The authors need to describe it.

Response 3: The description about primer for S100A8 was missing and has been added (line 156 to 160).

Point 4: Although line 195 to 196, you mentioned “In addition, among the S100 protein family members, S100A8 and S100A9 expression was specifically induced by IR injury in the lungs (Fig. 1C)”, what does “specifically” mean? If this indicate “in an idiosyncratic way”, it is out of line. The authors need to modify it.

Response 4: We appreciated the reviewer’s comment. We agree with the reviewer’s comment.  We used “particularly highly” instead of “specifically” (line 199).

Point 5: In Figure 1B, What does a8 or a9 mean? The authors need to modify Figure 1 and its legend.

Response 5: In order to clarify a8 and a9, we added a sentence in the legend of Figure 1B.

Point 6: The authors need to change the graph of S100a9 and S100a8 in Figure 1D. I think the numerical order is easier to read.

Response 6: We agree with the reviewer’s comment and changed the figure 1D.

Point 7: At line 269 and Figure 5A, you mentioned “indicating that anti-S100A8/A9 269 mAb improves lung function after IR”. If you use the word “improvement”, you should show the data when it got worse.

Response 7: The lung function got worse by IR (The IR-IgG group compared to the Sham groups), and this phenomenon was improved by S100A8/A9 mAb. 

Point 8: The immunohistograms in Figures 5 and 7 are difficult to interpret. The authors need to modify them including their legends.

Response 8: We modified Figure 5 and 7.

Point 9: Although this manuscript abruptly showed the results of an apoptotic experiments, but I could not understand the purposefulness of carrying out these experiments. In this regard, the authors should discuss the relationship between S100A8/S100A9 functions and necrosis, apoptosis including necroptosis and pyroptosis as mechanisms of cell death in acute lung reperfusion in the discussion section.

Response 9: We added and revised sentences about apoptosis in the Discussion (line 412-417).